# Do Single-Nucleotide Polymorphisms Affect Pain Intensity and Sufentanil Analgesia After Pediatric Scoliosis Correction Surgery?

**DOI:** 10.3390/ijms26083504

**Published:** 2025-04-09

**Authors:** Aleksander Turczynowicz, Jakub Równy, Weronika Przontka, Magdalena Grzesik, Piotr Jakubów, Oksana Kowalczuk

**Affiliations:** 1Department of Anesthesiology and Intensive Care for Children and Adolescents with Postoperative and Pain Treatment Unit, Medical University of Bialystok, 15-274 Bialystok, Polandjakubowpiotr@wp.pl (P.J.); 2Institute of Psychology, University of Lodz, 90-128 Lodz, Poland; 3Department of Clinical Molecular Biology, Medical University of Bialystok, 15-269 Bialystok, Poland

**Keywords:** SNP, single-nucleotide polymorphism, sufentanil, postoperative pain, opioid metabolism, OPRM1, COMT, ADRB1, pain intensity

## Abstract

Pain management in children remains a challenge. Postoperative pain assessment, which currently relies on behavioral and subjective scales, could be enhanced by the identification of single nucleotide polymorphisms effect on pain thresholds and opioid metabolism. This study explores the impact of nine SNPs—rs1799971, rs4680, rs4633, rs6269, rs4818 (with catechol-o-methyltransferase haplotypes), rs7832704, rs1801253, and rs1045642—on postoperative pain intensity, opioid requirements, coanalgesic use, C-reactive protein levels, and post-anesthesia care unit length of stay. This study involved 42 pediatric patients undergoing scoliosis correction surgery with postoperative sufentanil infusion. The genotyping was performed using real-time PCR with peripheral blood samples. Patients with the rs1801253 *ADRB1* GG genotype showed significantly lower 24 h NRS pain ratings (*p* = 0.032) and lower sufentanil infusion rates at the level of statistical tendency (*p* = 0.093). Patients with the rs1205 *CRP* CT genotype had a shorter PACU length of stay (*p* = 0.012). In contrast, those with the rs1045642 *ABCB1* GG genotype had a longer PACU stay by 0.72 h (*p* = 0.046). No significant associations were found for *OPRM1* rs1799971, *COMT*, or *ENPP2* SNPs. *ADRB1* rs1801253may be a novel SNP indicating higher postoperative pain risk, while rs1205 *CRP* and rs1045642 *ABCB1* could predict increased care requirements in PACUs. The *ADRB1* rs1801253 SNP may also predict opioid demand. These results suggest SNPs should be considered in acute pain assessment.

## 1. Introduction

Pain treatment in children is a challenging part of clinical practice due to difficulties in accurately assessing pain based on patients’ subjective reports. This is often affected by language skills, as well as cognitive and emotional immaturity, leading to incorrect postoperative pain intensity estimates and insufficient pain control [1,2,3].

Currently, behavioral scales such as NRS (Numerical Rating Scale), VAS (Visual Analogue Scale), Wong–Baker, and FLACC (Face, Legs, Activity, Consolability, Cry) are used to assess pain, especially in non-verbal patients. However, these scales have limitations since pain assessment is highly subjective [4,5,6].

In the case of postoperative pain, insufficient treatment may result in long-term negative outcomes from surgical procedures, such as chronic post-surgical pain. It is estimated that 11% to 13% of patients under the age of 18 experience chronic post-surgical pain [7,8,9].

Long-term studies on unoperated idiopathic scoliosis indicate the progression of disability and chronic pain. In pediatric patients, the percentage of lower back pain after surgery ranges from 7% to 77%. Even after several years, many patients still report back pain [10,11,12,13,14].

Patients undergoing scoliosis correction surgery are at high risk of developing chronic post-surgical pain. This extensive surgery causes significant tissue damage and requires postoperative pain treatment with highly potent opioids—fentanyl, remifentanil, and sufentanil [15,16].

Postoperatively, patients may not report any symptoms, but neurological exams often show impaired consciousness, raising the question of whether this is due to sufficient analgesic effect or excessive opioid sedation.

A challenge in pediatric pain therapy stems from differences in metabolism and body composition. Faster metabolic rate and a large volume of distribution due to a higher water content in children’s bodies significantly affect the pharmacokinetics and pharmacodynamics of opioids, the primary medication for intense postoperative pain [17].

In the scientific literature, research has focused on single-nucleotide polymorphisms (SNPs) and their effects on opioid metabolism, pain threshold, and pain intensity in both chronic and postoperative pain [18,19,20].

Commonly studied SNPs include those in the *mu-opioid receptor* gene rs1799971 A118G, in which the G allele is associated with reduced analgesic response to morphine due to altered μ-opioid receptor function. Catechol-O-methyltransferase (COMT) is an enzyme regulating epinephrine, norepinephrine, and dopamine degradation, which are involved in the modulation of pain pathways. Four SNPs—rs4680, rs4633, rs6269, and rs4818—located in the *COMT* gene affect enzyme thermostability and function. The *ATP-binding cassette 1* gene and *ABCB1* C3435T SNP have established significance for opioid metabolism, as the P-glycoprotein encoded by *ABCB1* is responsible for the transmembrane transport of opioids [21,22,23].

Recent studies have also investigated other SNPs related to opioids and pain sensation, such as *ADRB1* rs1801253 (beta-adrenergic receptor 1) rs1801253 causes a Gly389Arg amino acid substitution at the intracellular *C*-terminus of the β1-adrenergic receptor. It has been linked to variability in pain perception and analgesic effects of opioids such as fentanyl and remifentanil in Asian populations [24,25,26].

*ENPP2* encoding autotaxin, or ectonucleotide pyrophosphatase/phosphodiesterase family member 2, plays a role in nociceptive pain through inflammation, along with the *C-reactive protein* (CRP) SNP, which might affect postoperative pain through nociceptive pathways [27,28,29].

Most of these studies concern adults; however, significantly fewer studies focus on children [16,30,31,32]. Therefore, this article explores the impact of nine different SNPs (rs1799971, rs4680, rs4633, rs6269, rs4818, (as well as *COMT* haplotypes), rs7832704, rs1801253, rs1045642, and rs1205) on postoperative pain intensity, opioid doses, patient pain ratings, inflammation (measured by C-reactive protein levels), PACU (post-anesthesia care unit) length of stay, and coanalgesic use in pediatric patients (0–18 years old) undergoing elective scoliosis correction surgery. The goal is to determine if these SNPs could contribute to personalized therapy for acute and inflammatory pain in children, potentially improving recovery and reducing the risk of chronic pain or opioid dependence [33,34,35].

## 2. Results

### 2.1. Descriptive Statistics of Analyzed Clinical Parameters and Demographic Data

Forty-seven patients were enrolled in the study after obtaining signed informed consent. Due to the sampling error (n = 3) and missing samples due to laboratory error (n = 2), the final group included in the analysis comprised 42 patients. In the analyzed sample of N = 42 patients, there was a significant overrepresentation of females, accounting for 76.2% (n = 32) of the cohort, while males constituted a mere 23.8% (n = 10). The mean age of the participants was 13.33 years, with a standard deviation of 3.67 years, spanning from 5 to 18 years, suggesting a relatively narrow age distribution around adolescence. The mean weight of the patients was 47.72 kg, with standard deviation of 16.39, ranging from 16.5 kg to 82 kg. All demographic data are presented in Table 1, and descriptive statistics of the analyzed clinical parameters are shown in Table 2.

### 2.2. Influence of Confounders on Clinical Parameters

A regression-based approach was used to verify the influence of polymorphisms on clinical parameters. Linear regression models were applied to verify the influence of polymorphisms on numeric clinical parameters, while logistic regression was used to verify the association between polymorphisms and the usage of coanalgesics, as the usage of coanalgesics was described with a binary variable. Multivariate models were built to consider important confounders; however, given the total number of patients, each model included one polymorphism and one confounder as independent variables and a clinical parameter as the dependent variable. The considered confounders were age, gender, and type of procedure. Preliminarily, for each clinical parameter, its most important confounder was identified based on the lowest *p* value in the univariate models, and taking into account the condition of *p* < 0.157 (linear regressions) or *p* < 0.250 (logistic regressions) [36,37]. The outcomes of the univariate modelling between the confounders and clinical parameters are presented in Table 3. Age was selected as the most important confounder for CRP 72 h post-surgery. CRP 72 h post-surgery was higher by 6.75 in cases of patients older by one year (β = 6.75 CI95 [1.30;12.20]; *p* = 0.017). Gender was selected as the most important confounder for mean 24 h NRS pain rating; based on the lowest *p* value of 0.101, the outcome was not statistically significant, yet it suggested that females gave higher pain ratings. Age was selected as the most important confounder for sufentanil mean infusion flow rate (weight-adjusted). Procedure type was the most important confounder for PACU length of stay. PACU length of stay was shorter by 0.75 in cases of procedures other than Nuvasive or Expedium compared to Nuvasive (β = −0.75 CI95 [−1.34;−0.17]; *p* = 0.013). No confounder for coanalgesics fulfilled the condition of *p* < 0.250, and thus the impact of polymorphisms on the usage of coanalgesics was analyzed through univariate regression.

### 2.3. Influence of Polymorphisms on CRP 72 h Post-Surgery

A significant relationship was confirmed between rs4680 and CRP 72 h post-surgery. Patients with rs4680 AG had a higher CRP 72 h post-surgery level by 46.24 compared to patients with rs4680 AA (β = 46.24 CI95 [4.03;88.45]; *p* = 0.033). For graphic representation see Figure 1.

Additionally, relationships at the level of statistical tendency (*p* < 0.1) were found between *COMT* haplotype and CRP 72 h post-surgery. Patients with both ATCA/ATCA and *COMT* haplotypes other than ACCA/GTGG and ATCA/ATCA had lower CRP 72 h post-surgery levels compared to patients with ACCA/GTGG (β = −48.74 CI95 [−101.52;4.03], *p* = 0.069; β = −38.30 CI95 [−83.97;7.37], *p* = 0.098, respectively).

### 2.4. Influence of Polymorphisms on Mean 24 h NRS Pain Rating

A significant relationship was confirmed between rs1801253 *ADRB1* and mean 24 h NRS pain rating. Patients with rs1801253 *ADRB1* GG had a lower mean 24 h NRS pain rating by 1.92 compared to patients with rs1801253 *ADRB1* CC (β = −1.92 CI95 [−3.66;−0.17]; *p* = 0.032). For graphic representation see Figure 1. 

Additionally, a relationship at the level of statistical tendency (*p* < 0.1) was found between rs4818 and mean 24 h NRS pain rating. Patients with rs4818 CG had a lower mean 24 h NRS pain rating compared to patients with rs4818 CC (β = −0.64 CI95 [−1.40;0.11]; *p* = 0.091).

### 2.5. Influence of Polymorphisms on Sufentanil Mean Infusion Flow Rate (Weight-Adjusted)

No relationships with statistical significance were confirmed.

A relationship at the level of statistical tendency (*p* < 0.1) was found between rs1801253 *ADRB1* and sufentanil mean infusion flow rate (weight-adjusted). Patients with rs1801253 *ADRB1* GG had lower sufentanil mean infusion rate (weight-adjusted) compared to patients with rs1801253 *ADRB1* CC (β = −0.04 CI95 [−0.08;0.01]; *p* = 0.093).

### 2.6. Influence of Polymorphisms on PACU Length of Stay

A significant relationship was confirmed between rs1205 *CRP* and PACU length of stay, as well as between rs1045642 *ABCB1* and PACU length of stay. Patients with rs1205 *CRP* CT had a shorter PACU length of stay by 0.62 compared to patients with rs1205 *CRP* CC (β = −0.62 CI95 [−1.10;−0.14]; *p* = 0.012). Patients with rs1045642 *ABCB1* GG had a longer PACU length of stay by 0.72 compared to patients with rs1045642 *ABCB1* AA (β = 0.72 CI95 [0.01;1.43]; *p* = 0.046). For graphic representation see Figure 1. 

Additionally, a relationship at the level of statistical tendency (*p* < 0.1) was found between rs1045642 *ABCB1* and PACU length of stay. Patients with rs1045642 *ABCB1* AG had a longer PACU length of stay compared to patients with rs1045642 *ABCB1* AA (β = 0.51 CI95 [−0.08;1.09]; *p* = 0.087).

### 2.7. Influence of Polymorphisms on Usage of COANALGESICS

No relationships with statistical significance were confirmed.

Relationships at the level of statistical tendency (*p* < 0.1) were found between rs4818 and usage of coanalgesics, as well as between *COMT* haplotype and usage of coanalgesics. The odds of coanalgesic usage decreased by 70% in cases of patients with rs4818 CG compared to patients with rs4818 CC (OR = 0.30 CI95 [0.08;1.03]; *p* = 0.063). The odds of coanalgesic usage increased 3× in cases of patients with *COMT* haplotypes other than ACCA/GTGG and ATCA/ATCA, compared to patients with ACCA/GTGG (OR = 3.47 CI95 [0.82;16.32]; *p* = 0.099). See Table 4.

## 3. Discussion

Pain control after surgery in pediatric patients is still a major concern for healthcare providers [38]. Patients undergoing extensive surgeries like scoliosis correction are at risk of chronic postsurgical pain if acute pain is not properly treated. This can affect the surgery’s success and lower quality of life [35]. Pain assessment currently relies on behavioral scales, considering child developmental abilities, age, and gender. For instance, one of most important confounders for sufentanil dosage is female gender (at the level of statistical tendency) [39,40].

In this article, we explore the role of single-nucleotide polymorphisms (SNPs) in the complex interactions influencing pain perception and opioid use in the perioperative period [32,41]. The literature shows mixed results regarding SNPs’ ability to explain variation in pain perception and opioid requirements based on specific allele occurrences in different ethnic groups, e.g., Asian, European, or African [15,42,43,44].

To assess factors influencing clinical outcomes, we identified key confounders. Age, for example, influenced CRP levels post-surgery, likely due to the need for larger surgical incisions and increased spinal ossification in older patients [10]. Female gender confounded mean 24 h pain ratings without reaching statistical significance. However, this suggests that women report higher pain ratings after scoliosis correction. While female gender showed a tendency to influence 24 h pain ratings, it did not reach statistical significance. However, studies indicate that females generally experience higher postoperative pain and require more opioids [45,46,47]. Age was identified as the main factor influencing sufentanil dosage. Younger patients have a higher metabolic rate and larger distribution volumes, making weight-based opioid dosing alone insufficient [48].

We found a significant impact of rs4680 *COMT* on CRP levels after surgery, with higher levels in patients with the AG genotype. COMT is involved in inflammatory processes, and the A allele in the rs4680 SNP causes a non-synonymous substitution of valine to methionine at codon 158 (Val158Met), leading to changes in thermostability and variability in COMT enzymatic activity. The statistical tendency of haplotype influence on CRP levels also supports this explanation. This is a novel finding; however, it needs to be treated with caution as the study population is limited [49].

Our study also suggests that the *ADRB1* rs1801253 SNP influences pain ratings within 24 h of surgery in European patients. This finding is novel, as previous research has linked this SNP to remifentanil and fentanyl analgesic effects and pain intensity primarily in Asian populations. The most interesting finding is the fact that one study on 324 patients did not find any associations between this SNP and pain sensitivity and opioid requirements. Another study on 137 Chinese women revealed the *ADRB1* rs1801253 CC genotype to be associated with higher sensitivity to remifentanil analgesia [24,26]. The statistical tendency of lower pain intensity in patients carrying the CG genotype in the rs4818 SNP can be explained in the same way as for CRP levels, with the CG genotype potentially serving as a protective factor against postoperative pain [49,50,51]. Among European populations, G allele frequency is around 28%, and in Asian populations it is around 35% according to the ALPHA allele aggregator. This suggests that the results of studies regarding *ADRB1* are comparable among different ethnic populations, in contrast to *OPRM1* rs1799971, where the G allele in European populations is rare, thus giving no statistical effect on the investigated clinical parameters [52].

No statistically significant relationships between the SNPs and sufentanil dosage were found. However, there was a tendency for patients with the GG genotype of *ADRB1* rs1801253 to require less sufentanil post-surgery than patients with the CC genotype, which aligns with lower pain intensity. This is an interesting finding considering the fact that studies on Asian populations showed that the CC genotype required lower opioid doses for effective postoperative analgesia [24,26].

PACU length of stay was significantly affected by rs1205 *CRP* and *ABCB1* rs1045642. The shorter stay of the CT group of the rs1205 SNP might be predictive of lower pain due to a subsequent lower inflammatory response, and, furthermore, a reduced need for observation and therapeutic interventions regarding pain in the PACU. In contrast, the GG genotype of *ABCB1* may act as a predictor of higher needs in patients hospitalized in the PACU with regard to the need for effective postoperative pain control before discharge [29,53].

No significant influence of the SNPs was found on the use of coanalgesics like ketamine or lidocaine. Nevertheless, the rs4818 CG group showed a statistical tendency with 70% decreased odds of needing coanalgesics (*p* = 0.063), approaching the significance level of *p* < 0.05. This supports the literature suggesting that *COMT* variability impacts pain perception and that rs4818 CG might predict lower pain thresholds among patients undergoing major surgery [15,16].

No significant relationships were found with the *OPRM1* rs1799971 SNP, the most widely studied in pain and opioid research. This may be because the G allele, linked to higher pain and opioid needs, was rare in our group (7 out of 42, as expected for the European population), with only heterozygotes and no GG homozygotes [44].

As a result, the statistical analysis showed no influence on any clinical parameters of interest. Nonetheless, this SNP has the most substantial body of evidence, not only for its involvement in opioid metabolism, but also for its association with psychiatric disorders, such as alcohol and opioid addiction [54,55,56]. This is consistent with our previous findings on a smaller group of patients, as well as on larger cohorts of 126 and 207 European subjects, that this SNP may not predict pain thresholds or opioid needs in European populations [32,43,50].

## 4. Materials and Methods

### 4.1. Patient Population

This study was conducted at a clinical hospital, where children referred from lower reference centers nationwide were directed. Patients were admitted based on a referral order to the Orthopedics Department.

The cohort study ran from May 2022 to October 2024, involving patients scheduled for scoliosis correction surgery. The study followed the Declaration of Helsinki and received approval from the Bioethics Committee of the Medical University of Bialystok (consent no. APK.002.512.2021, 16 December 2021).

Inclusion criteria: children aged 0–18 years scheduled for scoliosis surgery, initial pediatric evaluation, and ASA (American Society of Anesthesiologists) I–II classification (assessed by an anesthesiologist with a legal guardian present). Eligible patients had no significant health issues besides scoliosis, or had stable, controlled symptoms.

Exclusion criteria: untreated or poorly controlled neurological disorders; neuropathic pain requiring opioids within three months prior to surgery; pulmonary conditions identified through spirometry, particularly restrictive lung diseases; metabolic disorders; cardiological issues (including arrhythmias and untreated congenital heart defects); and psychiatric disorders (such as borderline, histrionic, and schizoid personality disorders, phobias, anxiety disorders, depressive disorders, atypical bipolar disorders, and suspected addictions).

Symptoms of psychiatric disorders were evaluated during routine pediatric and anesthetic evaluations, and further psychological consultations were provided if needed. Patients who did not meet the criteria were qualified to have alternative treatment options.

### 4.2. Anesthesia and Analgesia

Before surgery, patients received oral midazolam (0.2 mg/kg) as a premedication, along with a crystalloid infusion after establishing peripheral intravenous access.

Upon arrival in the operating theater, patient identity and medical records were verified. Vital signs, including ECG, etCO2, saturation, and oxygen measurements, were continuously monitored.

Each patient received two peripheral venous catheters and a radial artery cannula for invasive blood pressure monitoring. Anesthesia depth was monitored using the BiSpectral Index (BIS), and peripheral nerve potential measurements were used to evaluate nerve damage [11].

Anesthesia induction included propofol and rocuronium, with dosages based on body weight, followed by remifentanil infusion and intubation using a reinforced tube. Anesthesia was maintained using TIVA-TCI (Total Intravenous Anesthesia—Target Controlled Infusion) with propofol and remifentanil (without additional neuromuscular blockers), adjusting infusion rates based on BIS to maintain a range of 35–50 [57].

### 4.3. Postoperative Pain Treatment

Before the end of the surgery and transfer to the postoperative ward, a 0.1 μg/kg/h sufentanil infusion was started to maintain analgesia after the remifentanil infusion. After operation, all patients received 15 mg/kg paracetamol and 16 mg/kg metamizole. Pain was assessed using the NRS.

The sufentanil infusion rate at admission, as well as pain score after extubation, at discharge from the postoperative ward, and 24 h after surgery were recorded. From this, the weight-adjusted mean sufentanil infusion rate and mean 24 h NRS pain intensity were calculated for statistical analysis [4,6]. CRP levels were measured 72 h after surgery as an indicator of inflammatory pain. If severe pain or adverse effects (e.g., disorientation, excessive sedation, breathing problems) occurred, coanalgesics like lidocaine (1–1.5 mg/kg/h) or ketamine (0.2–0.3 mg/kg) were added to maintain effective sufentanil doses.

### 4.4. Genotyping Data and Laboratory Analysis

To perform laboratory analysis, genetic blood samples (1.2 mL EDTA tubes) were collected before surgery and blood transfusions. Every sample had to be cured by mixing with erythrocyte-lysing solution, cooled down to 4 °C, and centrifuged for 10 min at 2500 rpm in a standard laboratory centrifuge.

Genotyping was performed using Roche Diagnostics LightCycler^®^ 480 (Roche Diagnostics International Ltd., Rotkreuz, Switzerland) real-time PCR hybridization probes for two-sequence SNPs with ready-made assays (TaqMan™ SNP Genotyping Assay: rs1799971, rs9397685, rs1205, rs6269, rs7832704, and rs1801253; TaqMan™ Drug Metabolism Assay: rs4680, rs4818, rs4633, and rs1045642 from Applied-Biosystems™ (Foster City, CA, USA)) and TaqMan Genotyping Master Mix. All genotyping and post-genotyping analyses followed the manufacturers’ protocols.

### 4.5. Statistical Analysis

Statistical analysis was performed in R software (version R4.4.2). Clinical parameters were described with mean and standard deviation (SD) in cases of numeric parameters with normal distributions, median and interquartile range (IQR) in cases of numeric parameters with non-normal distributions, and with n (%) in cases of categorical parameters. Distribution normality was verified with the Shapiro–Wilk test, skewness, and kurtosis. The influence of polymorphisms on clinical parameters was assessed with regression methods to consider confounders. In cases of clinical parameters of numeric type, linear regression was used; in cases of clinical parameters of categorical type, logistic regression was employed. Regression models verified how each polymorphism impacted each clinical parameter. Models were built as follows: the dependent variable was clinical parameter; independent variables were polymorphism and best confounder (one confounder was included in each model due to n = 42 number of patients). Potential confounders were age, sex, and type of procedure. For each clinical parameter, the best confounder was selected based on the lowest *p* value from the univariate models between the clinical parameter and the confounders, assuming that the minimum cut-off for *p* value in linear regression was *p* < 0.157 and the minimum cut-off for *p* value in logistic regression was *p* < 0.250. All statistical tests assumed significance when *p* < 0.05. Associations with *p* < 0.1 were indicated as a statistical tendency.

## 5. Limitations of the Study

This study has some limitations. The sample size (n = 42) restricted the complexity of statistical models, limiting us to two-predictor models. Additionally, low counts in some analyses reduced statistical power, requiring cautious interpretation of the results. While this approach provided meaningful insights, larger studies are needed to explore more complex interactions.

Additionally, although the *p*-values received from the two-predictor regression models indicated significant associations, multiple-testing correction led to adjusted *p*-values that did not reach significance (See Appendix A). This suggests that while the observed effects are promising, replication in larger samples is needed to confirm their robustness.

One of the criteria of eligibility for the study was no previous psychiatric disorders, including anxiety disorders and personality disorders. A lack of control for postoperative anxiety, nonetheless, is a limitation of this study. This factor could also contribute to variability in the results. Another limitation is the potential for selection bias due to the non-randomized nature of the sample.

Ethnic variability in SNPs is a limitation not only of this study, but also of the use of polymorphisms as a universal tool for improving pain assessment, as allele frequencies differ across ethnic groups. Consequently, a given SNP may serve as a significant pain biomarker only within a specific ethnic population.

Despite these constraints, our findings on both levels of statistical significance and statistical tendencies (*p* < 0,1) provide valuable insights and highlight the need for further research with larger cohorts to validate and expand upon these results.

## Figures and Tables

**Figure 1 ijms-26-03504-f001:**
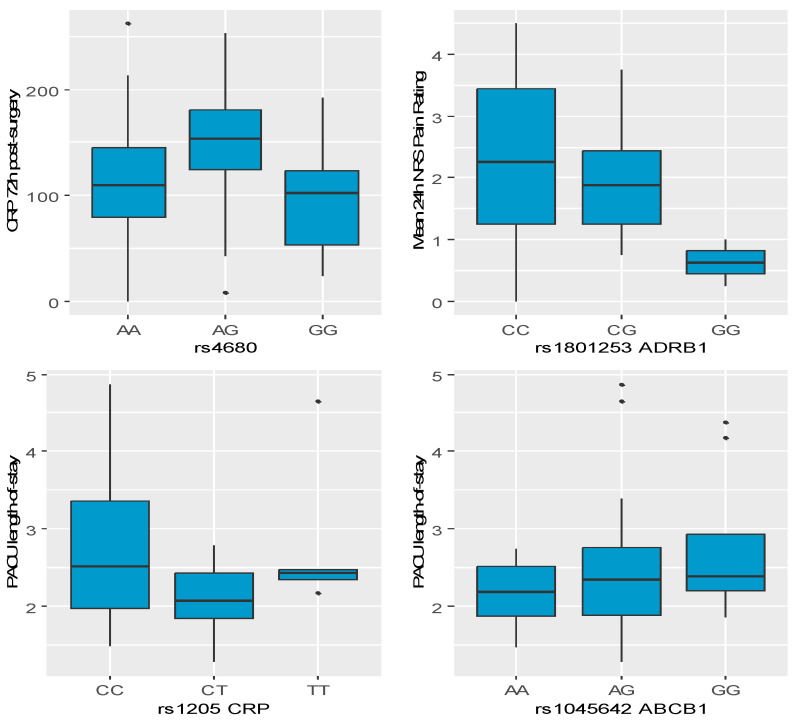
Boxplot charts presenting significant relationships between polymorphisms and clinical parameters based on regression modelling (Table 3).

**Table 1 ijms-26-03504-t001:** Demographic characteristics of the study group.

Variable	Mean ± SD/n (%)	Range
Age, years	13.33 ± 3.67	5.00–18.00
Sex, female	32 (76.2)	-
Weight, kg	47.72 ± 16.39	16.50 ± 82.00

SD—standard deviation.

**Table 2 ijms-26-03504-t002:** Descriptive statistics of analyzed clinical parameters by polymorphism groups.

Polymorphism		n	CRP 72 h Post-Surgery, Mean ± SD	Mean 24 h NRS Pain Rating, Mean ± SD	Sufentanil Mean Infusion Flow Rate (Weight-Adjusted), Mean ± SD	PACU Length of Stay, Median (IQR)	Coanalgesics, n (%)
rs1799971 OPRM1	AA	35	134.74 ± 66.90	2.09 ± 1.26	0.09 ± 0.03	2.38 (1.89;2.74)	17 (48.6)
	AG	7	99.97 ± 66.09	1.79 ± 1.19	0.08 ± 0.02	2.30 (2.03;2.37)	6 (85.7)
rs1205 CRP	CC	17	130.54 ± 64.63	1.91 ± 1.41	0.09 ± 0.04	2.52 (1.97;3.37)	10 (58.8)
	CT	20	121.22 ± 71.95	2.19 ± 1.19	0.09 ± 0.03	2.08 (1.85;2.42)	10 (50.0)
	TT	5	154.40 ± 62.81	1.85 ± 0.84	0.07 ± 0.03	2.43 (2.35;2.47)	3 (60.0)
rs1045642 ABCB1	AA	10	142.57 ± 72.25	2.42 ± 1.44	0.09 ± 0.03	2.18 (1.88;2.51)	6 (60.0)
	AG	23	129.80 ± 62.38	1.95 ± 1.27	0.09 ± 0.03	2.35 (1.89;2.76)	10 (43.5)
	GG	9	111.61 ± 77.68	1.83 ± 0.87	0.08 ± 0.02	2.38 (2.20;2.93)	7 (77.8)
rs6269	AA	13	132.59 ± 51.99	2.33 ± 1.04	0.09 ± 0.03	2.35 (1.98;2.50)	8 (61.5)
	AG	29	127.31 ± 73.86	1.91 ± 1.31	0.09 ± 0.03	2.30 (1.90;2.75)	15 (51.7)
rs4633	CC	6	92.05 ± 68.88	1.46 ± 1.73	0.08 ± 0.02	2.56 (2.10;2.88)	4 (66.7)
	CT	22	140.20 ± 66.49	2.15 ± 1.18	0.09 ± 0.03	2.35 (2.00;2.69)	13 (59.1)
	TT	14	127.06 ± 66.73	2.11 ± 1.10	0.09 ± 0.03	2.15 (1.85;2.48)	6 (42.9)
rs4818	CC	20	120.00 ± 59.27	2.42 ± 0.97	0.09 ± 0.03	2.32 (2.05;2.50)	14 (70.0)
	CG	22	137.07 ± 74.21	1.68 ± 1.36	0.09 ± 0.03	2.37 (1.89;2.89)	9 (40.9)
rs4680	AA	16	120.48 ± 67.53	2.09 ± 1.15	0.08 ± 0.03	2.29 (1.95;2.52)	8 (50.0)
	AG	20	145.01 ± 67.32	2.12 ± 1.19	0.09 ± 0.03	2.35 (1.95;2.76)	11 (55.0)
	GG	6	97.94 ± 61.44	1.58 ± 1.69	0.08 ± 0.02	2.19 (1.92;2.65)	4 (66.7)
COMT haplotype	ACCA/GTGG	13	160.77 ± 64.59	2.08 ± 1.26	0.09 ± 0.04	2.30 (1.88;2.52)	5 (38.5)
	ATCA/ATCA	10	119.55 ± 49.65	2.20 ± 1.09	0.09 ± 0.03	2.39 (2.00;2.51)	5 (50.0)
	Other	19	112.11 ± 72.17	1.92 ± 1.34	0.08 ± 0.02	2.30 (1.95;2.76)	13 (68.4)
rs7832704 ENPP2	AA	5	144.51 ± 44.08	2.40 ± 1.33	0.07 ± 0.03	2.38 (2.30;2.52)	3 (60.0)
	AG	37	126.84 ± 69.98	1.99 ± 1.23	0.09 ± 0.03	2.30 (1.88;2.73)	20 (54.1)
rs1801253 ADRB1	CC	26	134.91 ± 64.04	2.19 ± 1.35	0.09 ± 0.03	2.42 (2.02;2.77)	13 (50.0)
	CG	14	116.48 ± 78.04	1.95 ± 0.97	0.08 ± 0.02	2.02 (1.84;2.34)	9 (64.3)
	GG	2	138.62 ± 21.69	0.62 ± 0.53	0.06 ± 0.00	3.46 (3.10;3.82)	1 (50.0)

SD—standard deviation; IQR—interquartile range.

**Table 3 ijms-26-03504-t003:** Outcomes of univariate regression models created to indicate the most important confounder for each clinical variable.

Clinical Parameter	Age, Years		Gender, Female (vs. Male)		Procedure Type, Expedium (vs. Nuvasive)		Procedure Type, Other (vs. Nuvasive)	
	β/OR (95% CI)	*p*	β/OR (95% CI)	*p*	β/OR (95% CI)	*p*	β/OR (95% CI)	*p*
CRP 72 h post-surgery	6.75 (1.30;12.20)	** 0.017 **	19.28 (−30.19;68.76)	0.435	21.18 (−38.03;80.38)	0.474	−1.25 (−53.06;50.55)	0.961
Mean 24 h NRS pain rating	0.06 (−0.04;0.17)	0.249	0.74 (−0.15;1.62)	0.101	−0.46 (−1.54;0.63)	0.400	−0.25 (−1.19;0.70)	0.604
Sufentanil mean infusion flow rate (weight-adjusted)	0.00 (0.00;0.00)	0.060	0.02 (0.00;0.04)	0.084	−0.02 (−0.04;0.01)	0.207	0.00 (−0.03;0.02)	0.800
PACU length of stay	0.01 (−0.06;0.08)	0.718	−0.39 (−0.98;0.21)	0.195	0.02 (−0.65;0.69)	0.952	−0.75 (−1.34;−0.17)	** 0.013 **
Coanalgesics	0.94 (0.79;1.12)	0.514	1.29 (0.30;5.50)	0.729	0.89 (0.16;5.33)	0.892	0.44 (0.09;1.95)	0.288

β—model estimate from linear regression (all parameters except from coanalgesics); OR—odds ratio based on logistic regression (coanalgesics); CI—confidence interval. All outcomes were based on univariate regression models with clinical parameter as the dependent variable and a given confounder as the independent variable. The underlined confounders were chosen as the most important for each given clinical variable. No confounder was indicated for coanalgesics due to the condition of *p* < 0.250 not being met.

**Table 4 ijms-26-03504-t004:** Outcomes of multivariate regression models created to indicate the influence of polymorphism on clinical variables.

Polymorphism		CRP 72 h Post-Surgery		Mean 24 h NRS Pain Rating		Sufentanil Mean Infusion Flow Rate (Weight-Adjusted)		PACU Length of Stay		Coanalgesics (Yes/No)	
		β (95% CI)	*p*	β (95% CI)	*p*	β (95% CI)	*p*	β (95% CI)	*p*	OR (95% CI)	*p*
rs1799971 OPRM1	AA	ref		ref		ref		ref		ref	
	AG	−32.12 (−84.85;20.61)	0.225	−0.26 (−1.28;0.76)	0.612	−0.01 (−0.03;0.01)	0.410	−0.28 (−0.95;0.39)	0.405	6.35 (0.95;126.67)	0.102
rs1205 CRP	CC	ref		ref		ref		ref		ref	
	CT	−9.76 (−53.00;33.49)	0.650	0.25 (−0.57;1.08)	0.542	0.00 (−0.02;0.02)	0.837	−0.62 (−1.10;−0.14)	**0.012**	0.70 (0.18;2.57)	0.592
	TT	0.67 (−69.15;70.49)	0.985	0.06 (−1.22;1.34)	0.929	−0.01 (−0.04;0.03)	0.660	0.03 (−0.71;0.77)	0.927	1.05 (0.14;9.61)	0.962
rs1045642 ABCB1	AA	ref		ref		ref		ref		ref	
	AG	−12.66 (−62.23;36.91)	0.608	−0.37 (−1.33;0.58)	0.433	−0.01 (−0.03;0.01)	0.433	0.51 (−0.08;1.09)	0.087	0.51 (0.11;2.29)	0.386
	GG	−17.05 (−78.41;44.31)	0.577	−0.44 (−1.60;0.72)	0.451	−0.02 (−0.05;0.01)	0.117	0.72 (0.01;1.43)	**0.046**	2.33 (0.33;21.56)	0.410
rs6269	AA	ref		ref		ref		ref		ref	
	AG	9.41 (−35.39;54.20)	0.673	−0.27 (−1.12;0.58)	0.529	−0.01 (−0.03;0.01)	0.527	0.05 (−0.50;0.61)	0.843	0.67 (0.17;2.50)	0.556
rs4633	CC	ref		ref		ref		ref		ref	
	CT	48.04 (−9.93;106.01)	0.102	0.68 (−0.46;1.81)	0.234	0.01 (−0.02;0.03)	0.638	0.02 (−0.72;0.76)	0.963	0.72 (0.09;4.58)	0.737
	TT	20.56 (−41.83;82.95)	0.509	0.45 (−0.77;1.68)	0.461	0.01 (−0.02;0.04)	0.372	−0.21 (−0.99;0.57)	0.590	0.38 (0.04;2.61)	0.337
rs4818	CC	ref		ref		ref		ref		ref	
	CG	18.59 (−21.05;58.22)	0.349	−0.64 (−1.40;0.11)	0.091	0.00 (−0.02;0.02)	0.941	0.21 (−0.28;0.70)	0.395	0.30 (0.08;1.03)	0.063
rs4680	AA	ref		ref		ref		ref		ref	
	AG	46.24 (4.03;88.45)	**0.033**	0.34 (−0.55;1.22)	0.447	0.00 (−0.02;0.02)	0.830	−0.03 (−0.59;0.53)	0.914	1.22 (0.33;4.65)	0.765
	GG	−21.04 (−78.38;36.31)	0.462	−0.42 (−1.59;0.76)	0.479	0.00 (−0.03;0.03)	0.905	0.00 (−0.78;0.79)	0.991	2.00 (0.30;17.59)	0.488
COMT haplotype	ACCA/GTGG	ref		ref		ref		ref		ref	
	ATCA/ATCA	−48.74 (−101.52;4.03)	0.069	−0.10 (−1.18;0.99)	0.857	0.00 (−0.03;0.02)	0.746	0.06 (−0.62;0.75)	0.856	1.60 (0.30;8.90)	0.581
	Other	−38.30 (−83.97;7.37)	0.098	−0.29 (−1.20;0.62)	0.523	−0.02 (−0.04;0.00)	0.115	0.02 (−0.56;0.61)	0.932	3.47 (0.82;16.32)	0.099
rs7832704 ENPP2	AA	ref		ref		ref		ref		ref	
	AG	−1.93 (−65.13;61.28)	0.951	−0.56 (−1.74;0.62)	0.342	0.02 (−0.01;0.05)	0.234	−0.26 (−1.01;0.49)	0.488	0.78 (0.09;5.27)	0.802
rs1801253 ADRB1	CC	ref		ref		ref		ref		ref	
	CG	−20.26 (−63.27;22.75)	0.346	−0.53 (−1.34;0.29)	0.199	−0.01 (−0.03;0.01)	0.521	−0.15 (−0.69;0.39)	0.581	1.80 (0.48;7.26)	0.389
	GG	12.78 (−82.64;108.20)	0.788	−1.92 (−3.66;−0.17)	**0.032**	−0.04 (−0.08;0.01)	0.093	0.86 (−0.30;2.02)	0.140	1.00 (0.04;27.14)	>0.999

β—model estimate from linear regression (all clinical parameters except from coanalgesics) expressing the relationship between polymorphism and clinical parameter; OR—odds ratio based on logistic regression (coanalgesics) expressing the relationship between polymorphism and usage of coanalgesics; CI—confidence interval. All models assumed clinical parameter as a dependent variable and polymorphism as an independent variable, with additional confounders treated as independent variables. Age was a confounder for CRP 72 h post-surgery and sufentanil mean infusion flow rate (weight-adjusted), sex was a confounder for mean 24 h NRS pain rating, and type of procedure was a confounder for PACU length of stay. Models for coanalgesics do not include any confounders due to all potential confounders having *p* > 0.250 in the univariate analysis (Table 3).

## Data Availability

The data presented in this study are available on request from the corresponding author due to patient privacy reasons.

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
