# Peer review of "Do Single-Nucleotide Polymorphisms Affect Pain Intensity and Sufentanil Analgesia After Pediatric Scoliosis Correction Surgery?"

_ijms, 2025, doi:10.3390/ijms26083504_

Round 1

Reviewer 1 Report

Comments and Suggestions for Authors

1. In the abstract, more details are required particularly with background, methods, and conclusion. Please state a background outlining the need for assessing the SNPs relevant to this study. Explain all the acronyms when they appear first time.

2. In the introduction section, it is not clear on the roles of each of the SNPs relating to pain pathway. Please state the known and unknown associations of SNPs with pain/analgesia. Cite prior studies relevant to each SNP, their prevalence, observed effect sizes in relation to analgesic effects, including clinically relevant parameters.

3. For NRS, state median (range) instead of mean (SD) as they are not normally distributed as evident from the values in the Table 1. 

4. The sample size is very small for most of the SNPs. How to ensure that your study is adequately powered to find out the effect size between the SNPs? This is further compounded by concomitant analgesics.

5. Considering the potential impact of several variables on pain perception, a multivariate analysis should be done encompassing all the variables together and identifying the significant predictor of the outcome.

6. In fact, this reviewer suggest including the SNPs as one of the key predictor variables along with others and do regression tests for identifying the significance of genetic predictors of pain.

7. Having repeated statistical tests for so many times, how was the type 1 error (5%) controlled? Please use correction techniques such as Bonferroni and identify which of the variables are significantly associated with the outcomes following the corrected p-values. In the present version, there are a lot of chances for type 1 error.

8. Is there any error in the section 2.1? I can't see any text portion including the summary of patient demographics.

9. Please provide more details on the genotyping methods

Comments on the Quality of English Language

The language needs attention.

Author Response

  1. In the abstract, more details are required particularly with background, methods, and conclusion. Please state a background outlining the need for assessing the SNPs relevant to this study. Explain all the acronyms when they appear first time.

Thank you for your comment. We provided more details regarding background, methods and conclusions. All acronyms were checked and explained when appearing first time.

  1. In the introduction section, it is not clear on the roles of each of the SNPs relating to pain pathway. Please state the known and unknown associations of SNPs with pain/analgesia. Cite prior studies relevant to each SNP, their prevalence, observed effect sizes in relation to analgesic effects, including clinically relevant parameters.

Thank you. We explained the associations of SNPs with pain pathways or opioid metabolism as they appear in Introduction section with adequate studies cited. More explicit details regarding prevalence, effect sizes, analgesic effects and other clinical parameters can be found in the Discussion section. Prevalence of SNP genotypes is also discussed broader in Limitations section, as genotypes prevalence differ among ethnic populations, which can give different results of SNP influence on clinical parameters such as pain treshold, opioid requirement and tolerance depending on ethnicity of patient population included in the study.

  1. For NRS, state median (range) instead of mean (SD) as they are not normally distributed as evident from the values in the Table 1. 

Mean (SD) or median (IQR) as statistics for numeric variables were presented based on verification of normality. Distribution normality was verified with Shapiro-Wilk test, skewness and kurtosis. If Shapiro-Wilk test indicated normality, the distribution was treated as normal. If Shapiro-Wilk test did not indicate normality, the distribution was treated as normal only in cases when skewness and kurtosis suggested normality (skewness within range [-1;1] and kurtosis in range [2;4]). Verification of NRS for each polymorphism in attached table presents normality in all cases except from rs1799971 OPRM1 AA, where p=0,041 (Shapiro-Wilk), while the distribution was symmetric (S=0.31), which can be treated as close to normality and rs1801253 ADRB1 GG which consisted of n=2 observations. Given the outcomes of presented normality verification, we presented statistics of NRS with mean (SD)." 

  1. The sample size is very small for most of the SNPs. How to ensure that your study is adequately powered to find out the effect size between the SNPs? This is further compounded by concomitant analgesics.

Thank you for raising this point. Indeed, the number of patients in the study is limited. To account for this, we have considered the following:

- In each multivariate regression analysis which aimed to assess the associations between clinical parameters and polymorphisms we included only two predictors: one polymorphism and one confounder (identified in the first step of the analysis). In this way we followed the rule to have at least 20 events-per-variable, while some common approaches indicate the minimum of 15 events-per-variable (as e.g. in “Variable selection - A review and recommendations for the practicing statistician”, Variable selection - A review and recommendations for the practicing statistician - PubMed).

- We have assessed the power of each multivariate linear regression using pwr.f2.test function from pwr R package with the outcomes ranging from 0.67 to 0.97 for PACU length-of-stay, from 0.38 to 0.66 for Sufentanil mean infusion flow rate, from 0.32 to 0.77 for NRS and from 0.60 to 0.95 for CRP. For logistic regression models the same R function was used along with Negelkerky R2 to keep consistency of approach, which was proxy for power in logistic regression. The power for models with Coanalgetics varied highly from 0.06 to 0.61. Overall, power demonstrated variation with some outcomes properly powered, while other outcomes should be treated with caution. We additionally stated in discussion that some of the outcomes have lower power which is a limitation of the study.

However we agree that the sample size is small , however it was difficult to include larger group of patients due limit of patients that can be operated in our hospital’s Orthopedic Department. Therefore we conducted statistical analysis on the sample present, taking into account that the statistical power might not be high.

  1. Considering the potential impact of several variables on pain perception, a multivariate analysis should be done encompassing all the variables together and identifying the significant predictor of the outcome.

We are fully aware of the fact that there are at least several predictors impacting pain level, however given the number of patients in the study we decided to follow the two step procedure of (1) choice of best confounder for each analysed clinical parameter which were our dependent variables - out of age, sex and type of procedure and (2) for each clinical parameter we ran 10 models, one for each polymorphism, adjusting for the identified confounder. In this way we kept 2 independent variables in each model which we believed was the maximum number of predictors per model given the number of patients. Also, increasing the number of predictors would lead to lowered power. At the same time, we are aware that adding more predictors might increase the estimation accuracy, thus we added respective note in the limitations part.

  1. In fact, this reviewer suggest including the SNPs as one of the key predictor variables along with others and do regression tests for identifying the significance of genetic predictors of pain.

Thank you for your insight. We believe that we addressed your concern explicitly in previous answer. 

  1. Having repeated statistical tests for so many times, how was the type 1 error (5%) controlled? Please use correction techniques such as Bonferroni and identify which of the variables are significantly associated with the outcomes following the corrected p-values. In the present version, there are a lot of chances for type 1 error.

Thank you for your comment regarding multiple comparisons and the control of type I error. We acknowledge the concern and have applied the Benjamini-Hochberg (BH) correction due to higher power than Bonferroni method as well as balancing type I error control with statistical power. (“Why, When and How to Adjust Your P Values?”, Why, When and How to Adjust Your P Values? - PMC).

The uncorrected p values indicated significant associations, but the adjusted p values did not indicate significant outcomes. Thus, we would like to highlight potential associations based on unadjusted p values accompanied with confidence intervals for effect sizes, given the exploratory nature of our research. We added full outcome of the corrected p values in table 3a in appendix. The findings of the study should be interpreted with caution due to accomplished p-values and we have explicitly stated this in the limitations part of the discussion.

  1. Is there any error in the section 2.1? I can't see any text portion including the summary of patient demographics.

Thank you. Summary of patient demographics has been added do Materials and methods section of the manuscript.

  1. Please provide more details on the genotyping methods

More details are provided in section 2.4 of the manuscript regarding genotyping methods.

Reviewer 2 Report

Comments and Suggestions for Authors

Break the longest sentences into shorter, clearer sentences.

Introduction 

Introduction 

  • The introduction have to be restructured starting from  Pain assessment challenges in children, passing throughout Consequences of poor pain management (chronic pain, opioid effects, sedation vs. analgesia confusion), then Influence of metabolism and pharmacokinetics and Role of genetic polymorphisms in pain perception and opioid response and Endine with Rationale for the study and its objectives.

    Materials and Methods section have to be moved from the end of manuscript to its common collocation between introduction and results sections

    Results

    Consider structuring the text to first summarize univariate findings, then describe multivariate adjustments separately.

    While p-values are reported, effect sizes (β or OR) should be contextualized.

    Discussion

    Add a brief discussion on why this SNP may behave differently in European vs. Asian populations (e.g., genetic background, environmental factors, or sample size differences).

    Some cited limitations (sample size, ethnic variability) should be explicitly stated in a "Limitations" paragraph.

    In this paragraph add also these other two limitations: lack of control for psychological factors (e.g., anxiety, previous pain experiences) and potential bias due to non-randomized sample selection.

    References Appropriate 

    Tables and figures good quality. 

    •  
Comments on the Quality of English Language

None

Author Response

Break the longest sentences into shorter, clearer sentences.

Thank you for your comment. English editing was applied to the manuscript, long sentences have been edited to make them shorter and easier to read.

Introduction 

The introduction have to be restructured starting from  Pain assessment challenges in children, passing throughout Consequences of poor pain management (chronic pain, opioid effects, sedation vs. analgesia confusion), then Influence of metabolism and pharmacokinetics and Role of genetic polymorphisms in pain perception and opioid response and Endine with Rationale for the study and its objectives.

Materials and Methods section have to be moved from the end of manuscript to its common collocation between introduction and results sections

Results

Consider structuring the text to first summarize univariate findings, then describe multivariate adjustments separately.

In the first step we presented the outcomes of univariate modelling based on numbers in table 2, which aimed to identify the best confounder impacting each clinical variable. This was followed by presentation of outcomes from multivariate models (table 3) which aimed to understand how polymorphisms impact clinical parameters, adjusting for the confounder identified in the first step. We clarified the outcomes in table 3 by adding “multivariate models” into the table title as well as we did add notification under table 3 which described how the multivariate models were built

While p-values are reported, effect sizes (β or OR) should be contextualized.

With regards to effect sizes, we presented the interpretation of the outcomes in the Results section. We indicated how β or OR should be interpreted and what was the strength and direction of polymorphisms’ impact on clinical parameters.

Discussion

Add a brief discussion on why this SNP may behave differently in European vs. Asian populations (e.g., genetic background, environmental factors, or sample size differences).

Discussion has been extended to address ethnic factors contributing to different results among European, Asian and other populations when studying SNP influence on pain and opioid metabolism

Some cited limitations (sample size, ethnic variability) should be explicitly stated in a "Limitations" paragraph.

Sample size limitation has been stated explicitly in Limitations paragraph. We expanded ethnic variability limitations in Discussion section regarding all SNPs of interest as well as included in Limitations paragraph.

In this paragraph add also these other two limitations: lack of control for psychological factors (e.g., anxiety, previous pain experiences) and potential bias due to non-randomized sample selection.

Thank you, Limitations paragraph was created and all limitations of concern were addressed.

References Appropriate 

Tables and figures good quality. 

Thank you for your comment. References extended to support additions to discussion section. 

Round 2

Reviewer 1 Report

Comments and Suggestions for Authors

Thanks for the revision 

Reviewer 2 Report

Comments and Suggestions for Authors

The manuscript was improved like as suggested. 

The introduction section was well-restructured. 

The methods section was inserted in right position.

Results section was improved.

Discussion section was improved and limitation's paragraph was supplemented like as suggested. 

The manuscript now is worthy to be considered for pubblication.